# Algorithmic Assurance: An Active Approach to Algorithmic Testing using Bayesian Optimisation

**Shivapratap Gopakumar, Sunil Gupta,**[*] **Santu Rana, Vu Nguyen, Svetha Venkatesh**
Centre for Pattern Recognition and Data Analytics
Deakin University, Geelong, Australia

## Abstract

We introduce algorithmic assurance, the problem of testing whether machine learning algorithms are conforming to their intended design goal. We address this problem by proposing an efficient framework for algorithmic testing. To provide assurance, we need to efficiently discover scenarios where an algorithm decision deviates maximally from its intended gold standard. We mathematically formulate this task as an optimisation problem of an expensive, black-box function. We use an active learning approach based on Bayesian optimisation to solve this optimisation problem. We extend this framework to algorithms with vector-valued outputs by making appropriate modification in Bayesian optimisation via the EXP3 algorithm. We theoretically analyse our methods for convergence. Using two real-world applications, we demonstrate the efficiency of our methods. The significance of our problem formulation and initial solutions is that it will serve as the foundation in assuring humans about machines making complex decisions.

## 1   Introduction

Supervised learning algorithms today serve as proxies for decision processes traditionally performed by humans. As decision making processes get increasingly automated, it is reasonable to ask if our algorithms are behaving as intended. How far is the algorithm from the gold standard (human decision maker) it is serving as a proxy for? For example, consider a metallurgist who routinely makes decisions about elemental compositions to design a target alloy. If an algorithm is built to serve as a proxy for this decision process, can we provide assurance that the difference in the decision made by the algorithm and the metallurgist is within a stipulated bound? Similarly if an algorithm has been trained to recognize digits, can we ensure that the recognition error of the algorithm is acceptable across all allowable visual variations within which a human can recognise digits correctly? To provide such assurance we need to compare an algorithm against its gold standard and find the maximum deviation. An exhaustive comparison may solve this problem but would be prohibitively expensive as we need gold standard decisions for a large number of test instances. In absence of such a large set, how do we find such deviations efficiently?

Traditionally machine learning algorithms are tested by separating a small fraction of the available data as a validation set. Considering the validation set as a collection of random samples from the data space, we may need a large validation set to have high confidence on the algorithmic assurance, *i.e.* the maximal deviation of the algorithm from its gold standard is within an acceptable limit. Let us assume a hypervolume $v_\varepsilon$ wherein a function takes values within $1 - \varepsilon$ of its maximum. Then a random search will sample this hypervolume with the probability $\frac{v_\varepsilon}{V}$ where $V$ is the total search space volume. Assuming $V = R^d$ and $v_\varepsilon \approx r^d$, where $d$ is the input dimension, the random search scheme would need, on an average $\mathscr{O}\left( \left( \frac{r}{R} \right)^{-d} \right)$ number of samples [3]. This can be expensive - e.g.

---

[*]Corresponding author email:sunil.gupta@deakin.edu.au

if $\frac{r}{R} = 0.01$, nearly a million samples are required in just a three dimensional space. Therefore, a sample efficient alternative is needed for the algorithmic assurance.

We propose to use an active strategy for finding the maximum deviation that samples the data space such that each sample is only queried if it is aligned with the goal to find the maximum. Bayesian optimisation is one such efficient active learning method with a convergence guarantee on the average regret as $\mathscr{O}\left(\sqrt{\frac{d\ln T}{T}}\right)$ (using a Gaussian process model with squared-exponential kernel [4, 19]) where $T$ is the number of iterations/samples. Thus to reach the same regret level $\varepsilon$, Bayesian optimisation requires much smaller sample numbers. This approach actively recommends new instances during validation for which decisions are required from both algorithm and the human expert. Although costly, it remains practical because of the sample-efficiency guarantee of Bayesian optimisation. Our experience shows that it common to reach the maximum within tens of samples per dimension.

We develop a Bayesian Optimisation (BO) framework to efficiently discover the scenario wherein an algorithm maximally deviates from its gold standard. Given a difference function $y = f(x)$, where $x$ representing input instances and $y$ representing the difference of the algorithm's decision, our proposed algorithmic assurance framework aims to efficiently discover the instance for which the algorithmic decision differs most from the gold standard. We assume that functions underlying the decision making of both the gold standard process and the algorithm are smooth and therefore their difference function $f(x)$ is also smooth. We model $f(x)$ using a Gaussian process [15], and its predictive distribution is used to predict the deviation of the algorithm from the gold standard along with any epistemic uncertainty. This prediction is then used to construct a cheap surrogate function (acquisition function) that takes higher values at points where either the algorithm deviation or the epistemic uncertainty is high. The acquisition function is finally maximised to recommend a new instance for the algorithm testing. Both the gold standard and the algorithm decisions are then acquired to evaluate $f(x)$ at the new instance and this information is used to update the Gaussian process model of $f(x)$. This process iterates until convergence. We call this framework *single-task algorithmic assurance*.

We next move to *multi-task algorithmic assurance* where we extend our framework to provide assurance for algorithms that have vector-valued outputs. Our goal now is to find the scenario where an algorithm maximally deviates from its gold standard across *any* output. For example, in alloy design, elements are combined and heated, leading to phase formations. The strength of the resultant alloy is related to these phase fractions. An algorithm can be used to model the relation between the elemental composition and phases. Some phases are more common, thus statistics for each phase is not equally strong. This makes the rarer phase prediction more error prone. We therefore need to *efficiently* find the elemental composition where our algorithm's phase prediction maximally deviates from the true phase values across any phase, since predicting each phase is equally important. This boils down to a BO problem with $C$ black-box expensive functions and our task is find the largest *global maximum across all the functions*. To address this efficiently, we formulate each function as an arm of a multi-arm bandit and define the reward for pulling an arm as the best value found by BO for the corresponding function (up to any iteration). This method can efficiently switch across $C$ optimisation problems to quickly discover the optimum point. We theoretically analyse this algorithm and show that its simple regret has the order $\mathscr{O}\left(\sqrt{\frac{d\ln T}{T}} + \sqrt{\frac{C\ln C}{T}}\right)$.

It may appear superficially that the multi-task BO [20] is related, however, this method optimizes multiple related functions concurrently through mutual learning. We note that our problem is different in two ways: (1) we do not aim to maximise each function, rather quickly identify the function with the largest maximum and then find its maximum point; and, (2) multiple functions in our setting need not be related, which is a crucial assumption in the multi-task BO.

We demonstrate our framework on two problems: Prediction of strength-determining phases in an alloy design process, and recognition of handwritten digits under visual distortions. Our main contributions are:

- Introduction of a new notion of *algorithmic assurance* to assess the deviation behaviour of an algorithm from its intended use;
- Construction of an efficient framework for algorithmic assurance in both single and multi-task settings;

- Demonstration of the efficiency of our methods using two real world applications.

The significance of our problem formulation and solutions is that it will be the first step towards providing assurance to users of an algorithm.

## 2 An Active Approach to Algorithmic Testing

In this section we present our proposed framework for efficient algorithmic testing. Let us assume we have an unknown function $a(x)$ to be modelled using a set of observations of the form $D_n^{\text{train}} = \{(x_i^{\text{tr}}, o_i^{\text{tr}}), i = 1, \ldots, n\}$. Given the dataset $D_n^{\text{train}}$, a typical approach is to use a machine learning algorithm (e.g. a neural network) to learn an approximation $\mathscr{A}(x)$ of $a(x)$. Define a function $f(x) = \mathscr{L}(a(x), \mathscr{A}(x))$ that measures the deviation of $\mathscr{A}(x)$ from $a(x)$ at any point $x$. Various form of deviation can be used, for example, $\mathscr{L}(a(x), \mathscr{A}(x)) = (a(x) - \mathscr{A}(x))^2$ when dealing with a regression problem. In our proposed algorithmic testing framework, our goal is to efficiently identify a scenario $x^*$ wherein the algorithm output $\mathscr{A}(x^*)$ maximally deviates from the function $a(x^*)$. We express this goal through the following optimisation problem

$$x^* = \operatorname*{argmax}_{x \in \mathscr{X}} f(x) = \operatorname*{argmax}_{x \in \mathscr{X}} (a(x) - \mathscr{A}(x))^2 \tag{1}$$

Since function $f(x)$ is not known analytically, the objective function in the above optimisation problem is treated as a black-box function. In addition, evaluating $f(x)$ is expensive. The problem is thus finding the optimum of an expensive, black-box function.

**Bayesian Optimisation:** A method that has recently become popular for efficient global optimisation of expensive, black-box functions is Bayesian optimisation (BO) [17, 6, 7, 13]. It represents the black-box function through a probabilistic model, which is then used to reason about where in the space the optimum is located (for exploitation of available knowledge) and where we have the least knowledge about the function (for exploration for further knowledge). Based on this reasoning, the function is evaluated at a new location balancing the exploration and exploitation requirements and the new observation is used to update the function model. This sequential procedure repeats until the global optimum is reached or the optimisation budget is exceeded. The BO algorithms [19, 4] come with an efficiency guarantee on their convergence and usually have sub-linear growth rate for cumulative regret.

Gaussian processes are most popular for modelling the unknown function when doing Bayesian optimisation though other models have also been used [9, 18, 14]. Using Gaussian process prior, a function is modelled as $f(x) \sim \text{GP}(m(x), k(x, x'))$, where $m$ is a mean function and $k(x, x')$ contains the covariance of any two points on the function. With availability of noisy observations of the form $y_i = f(x_i) + \varepsilon_i$ (where $\varepsilon_i \sim \mathscr{N}(0, \sigma_\varepsilon^2)$), collectively denoted as $D_t = \{x_i, y_i\}_{i=1}^t$, we can derive the predictive distribution for the function value at a new observation $x'$ to be a Gaussian distribution [15] - its mean and variance are given as $\mu_t(x') = k^T(K + \sigma_\varepsilon^2 I)^{-1} \mathbf{y}$ and $\sigma_t^2(x') = k(x', x') - k^T(K + \sigma_\varepsilon^2 I)^{-1} k$ where assuming $k$ as a covariance function [15], $K$ is a matrix of size $t \times t$ whose $(i, j)$-th element is defined as $k(x_i, x_j)$ and $k$ is a vector (overloaded notation) with its $i$-th element defined as $k(x, x_i)$.

A nice property of BO when using Gaussian processes is that it usually avoids convergence to any "spurious peaks" and mostly converges to a stable peak. This property is useful for our algorithmic testing framework when we are interested in finding not just the location of the largest deviation of the algorithm but a region where the deviations are generally high. This may help in understanding the reasons of algorithm deviation and any potential remedies.

**An illustrative example:** To understand how BO avoids convergence to any "spurious peaks", let us consider an illustrative example function $f(x)$ with two peaks (see Figure 1) at locations $x_0$ and $x_0'$ such that $f(x_0) > f(x_0')$. Now consider two cases such that in the first case, the peak at $x$ is sharper (red) than the second case (grey). When using a Gaussian process model, we can show that if the two cases have previous observations at the same locations $\{x_1, \ldots, x_t\}$, the predictive mean of the Gaussian process model $\mu_t(x)$ for case-1 will be lower than that of case-2. This is because $\mu_t(x) = k^T(K + \sigma_\varepsilon^2 I)^{-1} \mathbf{y}$ and since $y_i$'s of the case-1 are only equal or lower than the corresponding $y_i$'s of case-2. With the assumption of previous observations for the two cases being at the same locations, the predictive variance of the Gaussian process model $\sigma_t(x)$ for both cases would be equal. Thus an acquisition function $\alpha_t(x)$ (based on typical acquisition function such as GP-UCB [19] or EI [10]) will take a lower value for case-1 than case-2. Since the two cases mainly differ around location $x_0$ (see Figure), the

acquisition function $\alpha_t(x)$ around $x_0$ will be lower for case-1 than case-2. Therefore the probability of a point $x$ around $x_0$ being the maxima of $\alpha_t(x)$ is lower for case-1 than case-2. Further, the narrower the peak of case-1, the lower is this probability. Therefore, Bayesian optimisation algorithm converges to the narrower peak with lower probability. This result would generally hold as long as the observations used in BO have a small measurement noise. If convergence to narrow peaks is becoming unavoidable or common, one may resort to BO methods that are customised to avoid spurious peaks [12, 5].

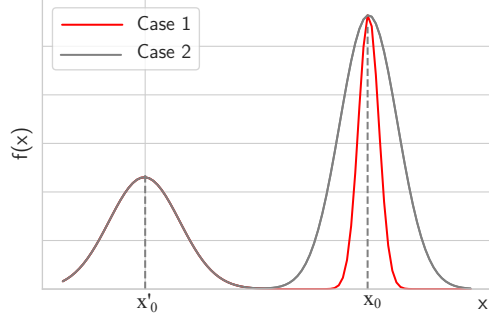

Figure 1 – An example function illustrating spurious (red) and wider (grey) peaks.

## 3 Multi-Task Algorithmic Testing

There are several applications where we need to model vector-valued outputs. In other words, this involves modelling multiple outputs or tasks. For example, in alloy design, for each composition of constituent elements, we have multiple phases. Let us assume that we have trained one machine learning model for each of these tasks. These models can be either independently or jointly trained depending on whether the tasks are independent or related. Since each task is different, the scenario where the algorithm output maximally deviates from the true output differs from task to task. Our above-mentioned single-task algorithmic testing method can be applied to this multi-task problem by aggregating the deviations for all tasks and thus can only discover the scenario where the algorithm deviates from the true output in an average sense. However, when it is important to get the assurance for each task or output, this approach may be insufficient.

In our proposed multi-task algorithmic testing, we aim to efficiently discover the scenario wherein the algorithm maximally differs from the true function for *any* of the outputs or tasks. Let us assume that there are $C$ tasks, indexed as $c = 1, \ldots, C$ and for $c$-th task, the true and the trained algorithm functions are $a_c(x)$ and $\mathscr{A}_c(x)$ respectively. We denote the discrepancy functions between the algorithm and the true functions by $f_1(x), \ldots, f_C(x)$. Each function has an optimum $f_c^* = \max_{\forall x \in \mathscr{X}} f_c(x)$. We aim to find both the optimal index $c^*$ such that $c^* = \operatorname{argmax}_{c \in C} f_c^*$ and the optimizer location $x^* = \arg\max_{x \in \mathscr{X}} f_{c^*}(x)$.

A simple approach to solve our problem is to perform Bayesian optimisation for each function $f_c(x)$ to obtain $f_c^*$ and then finally find $c^*$ and $x^*$. However this approach is inefficient as it unnecessarily evaluates the suboptimal functions for their complete Bayesian optimisation sequence. Our intuition is that it is possible to identify the tasks for which the algorithm has high errors within few function evaluations from all tasks and then mostly perform function evaluations for tasks with high deviations from the gold standard. In multi-arm bandit (MAB) research, this problem can be thought of identifying an arm with the best reward (or simply the "best arm"). There are several algorithms to identify the best arm, e.g. UCB1, $\varepsilon$-greedy, Hedge, EXP3 etc [1, 2]. Of these, Hedge and EXP3 are the algorithms that can be used under most general conditions with few assumptions on reward distributions unlike UCB1 and $\varepsilon$-greedy that require *i.i.d.* assumption. In our case, at any iteration of Bayesian optimisation, we define the reward of choosing a task at any iteration as the best function value reached up to that iteration from that task. Since the "best so far" statistics is not independent across iterations, the reward distribution is not *i.i.d.* The use of Hedge algorithm with BO has been considered earlier by [8] in a different context to ours. The Hedge algorithm in [8] is used to select acquisition functions for Bayesian optimisation. A requirement of the Hedge algorithm is that it needs the observation of rewards from each arm at all iterations. Unfortunately this requirement is not met in our scheme as if we only receive the reward for the selected arm – a *partial* reward feedback scenario. Therefore, for our multi-task algorithmic testing framework we use EXP3 algorithm [2, 16] which is capable of working in a partial reward feedback scenario.

Using the EXP3 algorithm we proceed as follows. At each iteration $t$, we first select a function indexed as $h_t = c$ and then advance its (one step) Bayesian optimisation to select the next point for evaluation by maximising the acquisition function as $x_t = \operatorname{argmax}_{x \in \mathscr{X}} \alpha_t(x \mid D_t(h_t))$ where $D_t(h_t)$ are observations up to iteration $t$ for the task indexed as $h_t$. The reward for the selected function is denoted

by $g_t(c)$ and is defined as the best function value so far, *i.e.* $g_t(c) = \max_{\forall x_i \in D_t(c)} f_c(x_i)$. Using rewards $g_t(c)$, we compute a probability $p_t^c$ as $p_t^c = (1-\eta)\frac{\omega_c}{\Sigma_{c=1}^C \omega_c} + \frac{\eta}{C}$, where $\omega_c = \omega_c \times \exp(\eta \hat{g}_t(c)/C)$ and $\eta = \sqrt{\frac{C \ln C}{(e-1)T}}$ is a EXP3 parameter pre-defined given the maximum budget $T$ (as per Corollary 3.2 of [2]). The probability vector $p_t = [p_t^1, ..., p_t^C]$ indicates the promise of different tasks for obtaining high values and is used to select a function for performing Bayesian optimisation. This process continues iteratively either until convergence or the function evaluation budget is exhausted. We refer to this algorithm as *EXP3BO* (see Algorithm 1).

---

**Algorithm 1** EXP3BO Algorithm for Multi-task Algorithmic Testing

---

**Input** $\eta = \sqrt{\frac{C \ln C}{(e-1)T}}$, $C$ #categorical choice, $T$ #max iteration

1: Init $\omega_c = 1, \forall c = 1...C$.
2: **for** $t = 1$ to $T$
3:       Compute the probability $p_t^c = (1-\eta)\frac{\omega_c}{\Sigma_{c=1}^C \omega_c} + \frac{\eta}{C}, \forall c = 1...C$.
4:       Choose a categorical variable at random $h_t \in [1, ..., C] \sim p_t = [p_t^1, ..., p_t^C]$.
5:       Optimize the acquisition function $x_t = \arg\max \alpha_t (x|D_t(h_t))$ given $h_t$.
6:       Evaluate the blackbox function $y_t = f([x_t, h_t = c])$ and augment $D_t(h_t) = D_{t-1}(h_t) \cup (x_t, y_t)$.
7:       Update the reward $g_t(h_t) = \max_{\forall x_i \in D_t(h_t)} f_{h_t}(x_i)$ and normalise as $\hat{g}_t(h_t) = g_t(h_t)/p_c^t$.
8:       Update the weight $\omega_{h_t} = \omega_{h_t} \times \exp(\eta \hat{g}_t(h_t)/C)$.
9: **end for**
**Output:** $\mathscr{D}_T$

---

### Convergence Analysis

We now present the convergence analysis. All the bounds are probabilistic bounds that hold with high probability. Let $\gamma_T$ be the maximum information gain over any $T$ iterations, it can be bounded for common kernels (e.g. for SE kernel $\gamma_T \sim O\left((\ln T)^{d+1}\right)$) [19].

**Lemma 1.** *(Due to [19]) Let $T$ be the number of iterations, $d$ be the input space dimension, then we can bound the simple regret $S_T$ after $T$ iterations of GP-UCB by a sublinear term as*

$$S_T = f_c^* - \max_{\forall x_t, t \leq T} f_c(x_t) \leq \frac{1}{T}\sum_{t=1}^T (f_c^* - f_c(x_t)) \leq \mathscr{O}\left(\sqrt{\gamma_T \ln T/T}\right).$$

Since we do not know which function among $f_1, ..., f_C$ has the overall maxima $f^*$, as discussed earlier a naïve algorithm can divide any available function evaluation budget $T$ equally among $C$ options. We refer to this algorithm as *Round-robin BO*. This algorithm only allocates $\frac{T}{C}$ evaluations for the optimal function indexed by $c^*$. We next provide the convergence rate for this Round-robin algorithm and later show that our proposed EXP3BO algorithm will have a tighter bound than the Round-robin BO. Another similar naïve approach (Random Categorical BO) is to randomly select a function and optimise. On average, this approach will also allocate $\frac{T}{C}$ evaluations for each function.

**Lemma 2.** *Given $C$ choices, the Round-robin BO and the Random Categorical BO methods will have the simple regret bounded as $S_T \leq \mathscr{O}(\sqrt{C\gamma_T \ln T/T})$.*

*Proof.* Since these methods allocate only $\frac{T}{C}$ evaluations to optimize the optimal function $f_{c^*}(x)$, using Lemma 1 we can write the simple regret bound as $S_T = S_{\frac{T}{C}}(c^*) \leq \mathscr{O}(\sqrt{\frac{C\gamma_T}{T} \ln \frac{T}{C}})$. We can see that the regret increases as $\mathscr{O}(\sqrt{C})$. $\qquad \square$

**Lemma 3.** *(Due to [2]) For $T > 0$, setting $\eta = \sqrt{\frac{C \ln C}{(e-1)T}}$, the expected regret of the EXP3 algorithm is bounded as*

$$\max_{h \in [C]} \sum_{t=1}^T g_t(h) - \mathbb{E}\left[\sum_{t=1}^T g_t(h_t)\right] \leq \mathscr{O}\left(\sqrt{TC \ln C}\right),$$

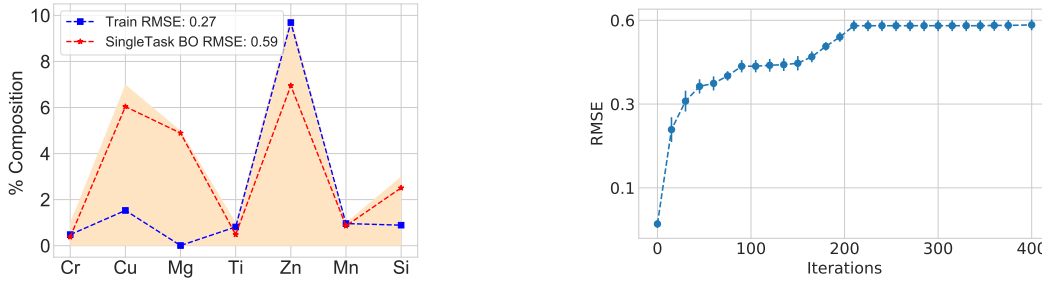

Figure 2 – *Left:* Element compositions (parallel coordinates) for max error during training and test. Bounds for each element composition in shaded regions. *Right:* Convergence of test error (RMSE).

*where we denote* $g_t(c) = \max_{\forall x_i \in D_t(c)} f_c(x_i)$. *The expectation is under randomness in the algorithm to select* $h_t$.

**Theorem 4.** *The EXP3BO algorithm has its simple regret bounded by*

$$\mathbb{E}\left[S_T^{Exp3BO}\right] \leq \mathcal{O}\left(\sqrt{\gamma_T \ln T / T} + \sqrt{C \ln C / T}\right).$$

*Proof.* Let $f^* = \max_{\forall c \in C, \forall x \in \mathcal{X}} f_c(x)$ be the optimum value that we seek. From Lemma 3, we can write

$$f^* - \mathbb{E}\left[\frac{1}{T}\sum_{t=1}^{T} g_t(h_t)\right] - \left\{f^* - \max_{h \in [C]} \frac{1}{T}\sum_{t=1}^{T} g_t(h)\right\} < \mathcal{O}\left(\sqrt{\frac{C \ln C}{T}}\right). \qquad (2)$$

Since $\frac{1}{T}\sum_{t=1}^{T} g_t(h_t) \leq g_T(h_T)$, we have $\mathbb{E}\left[S_T^{EXP3BO}\right] = f^* - \mathbb{E}[g_T(h_T)] \leq f^* - \mathbb{E}\left[\frac{1}{T}\sum_{t=1}^{T} g_t(h_t)\right]$. Denote the oracle simple regret as $S_T^{Oracle} = f^* - \max_{h \in [C]} \frac{1}{T}\sum_{t=1}^{T} g_t(h)$. Further assuming that the best arm can be identified by oracle with high probability, using Lemma 1, we have $S_T^{Oracle} \leq \mathcal{O}\left(\sqrt{\gamma_T \ln T / T}\right)$ and thus

$$\mathbb{E}\left[S_T^{EXP3BO}\right] < \mathcal{O}\left(\sqrt{C \ln C / T}\right) + \mathcal{O}\left(\sqrt{\gamma_T \ln T / T}\right).$$

$\square$

We can see that the regret bound remains sublinear in $T$ and is tighter than the regret bound of the Random Categorical or the Round-robin algorithm.

## 4 Experiments

We evaluate single and multi-task assurance using the two real world applications: (1) Alloy design, and (2) hand written digit recognition. In our algorithm, a squared exponential kernel is used for BO. All our results are reported by aggregating results from 10 runs with each run initialized randomly.

### 4.1 A neural network model predicting alloy-strengthening phases

Alloys are mixtures of elements that are able to achieve properties that are not possible by a single element. Laboriously collected experimental data elaborate how a mixture of elements form "phases". A phase is a homogeneous part of the alloy that has uniform physical and chemical characteristics, and determines the alloy strength. Experimental data for alloys are contained in proprietary simulators (eg. Thermocalc) and experimenters query such simulators for computed phase characteristics. These complex computations are expensive.

We construct a proxy algorithm for Thermocalc using a neural network to predict phases. We then apply our model to discover the test data point where the network prediction differs most from the Thermocalc output. Our proxy network is trained on 1000 samples generated from Thermocalc for

Aluminium 7000 series alloys that mainly consists of Aluminium and seven other elements (Cr, Cu, Mg, Ti, Zn, Mn, Si) whose % compositions are in a defined range as shown by the shaded region in Fig. 2 (*Left*). Input to the network is a 7 dimensional vector of element compositions. The output is a vector of alloy phases. After consulting with domain experts, we model 16 relevant phases. Our neural network consists of 2 hidden layers with 14 and 36 nodes respectively. A 30% dropout was introduced between the second layer and the output layer. The network was trained to minimize the error averaged over 16 phases. The neural network was trained for 100 epochs using a batch size of 5. The alloy composition corresponding to the maximal training RMSE of 0.27 was: Cr = 0.85 %, Cu= 2.06 %, Mg=0.18 %, Ti= 0.88 %, Zn= 8.25 %, Mn=0.37 %, and Si= 0.56 % (Fig. 2 (*Left*)). We use this neural network model for single and multi-task assurance. Single task measures the average error made across all phases, whilst multi-task measures error in each phase individually. In our notation, **x** denotes an elemental composition and **y** denotes the error in phase prediction.

### 4.1.1 Single task assurance

We run BO for our single task assurance to discover the composition with the maximal deviation from Thermocalc. The optimisation result is shown in Fig. 2 (*Right*). The element composition discovered by our method corresponding to the maximal error is Cr = 0.38 %, Cu = 6.04 %, Mg = 4.89 %, Ti = 0.48 %, Zn = 6.95 %, Mn = 0.86 % and Si = 2.51 %. As seen from Fig. 2, our algorithm discovers a significantly different composition with a much larger error (0.59) in just about 200 iterations.

### 4.1.2 Multi-task assurance

We use EXP3BO for multi-task assurance to discover the composition with the maximal deviation from Thermocalc for any alloy phase. Instead of measuring the error averaged across all phases, we consider the error of each phase separately. This gives rise 16 error functions where the highest error needs to be found efficiently without exhaustively optimising all. To evaluate the optimisation efficiency of our proposed EXP3BO algorithm we compare it against the baselines - Round-robin BO, Random Categorical BO and SMAC [9]. To find the phase that has the highest error (Oracle), we run BO for each phase separately and identify the phase with the highest error ($c^*$). Fig. 4 (*Left*) shows that the EXP3BO outperforms other baselines and reaches close to the Oracle. Also it accurately identifies the "AL12MN" phase that has the highest error - see histogram in Fig. 4 (*Right*). We

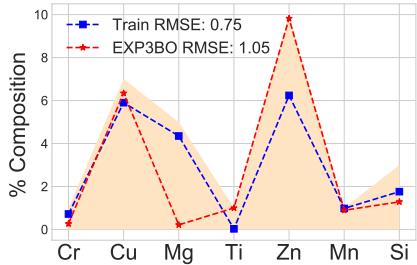

Figure 3 – Element compositions (parallel coordinates) for maximal error found during training and test stages by EXP3BO.

found the maximal error for "AL12MN" phase for RMSE=1.05, at a substantially different element composition compared to the one found during the algorithm training stage (see Fig. 3).

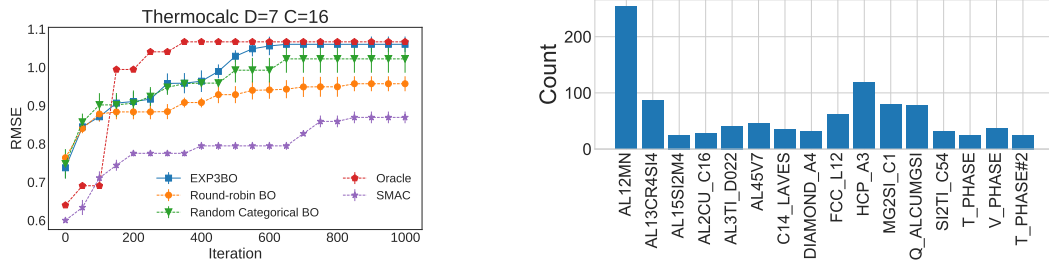

Figure 4 – Alloy phase prediction using EXP3BO. *Left*: Performance comparison - RMSE vs iterations. *Right*: Histogram of phases selected. It converges and exploits "AL12MN" phase more.

## 4.2 A convolutional neural network for handwritten digit recognition

We construct a proxy algorithm for recognising digits and the task is to identify the level of distortion causing the largest error in a transformed MNIST[11] dataset. In our notation, **x** denotes a visual

distortion (shear and rotation) while **y** denotes recognition error. The training data consists of MNIST images distorted with shear ($sh_x$, $sh_y$) and rotation ($\theta$). Our training dataset is created as follows: Each MNIST digit is first randomly sheared between $sh_x, sh_y \in [-0.2, 0.2]$, followed by a random rotation $\theta \in [0, 360]$. We removed digit 9 from our data to avoid confusions with digit 6 when subjected to rotation transform. $54,051$ such sheared and rotated MNIST digits are used for training a CNN. We use the LENET-5 architecture (as in [11]) with learning rate = $10^{-3}$ and number of epochs set to 20. The mean training error was found to be 4.1%. Maximal error from grid search was found to be 5.7% at shear ($sh_x = -0.2, sh_y = -0.2$) and rotation $\theta = 3°$.

### 4.2.1 Single assurance task

We run BO for our single task assurance to discover the distortion for maximal recognition error. The optimisation result is shown in Fig. 5. BO discovered a highest error of 7.1% at distortion parameters ($sh_x = 0.088, sh_y = -0.2$) and rotation $\theta = 175.7°$.

### 4.2.2 Multi-task assurance

We use EXP3BO for multi-task assurance to discover the maximal recognition error for any digits. Instead of measuring the error averaged across all digits, we consider error of each digit separately. This is important because the error in recognising each digit may differ depending on its visual complexity and distortion. Once again we compare EXP3BO with the baselines described in Section 4.1.2. The performance of EXP3BO is superior

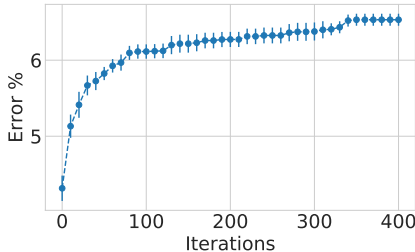

Figure 5 – Single task assurance for digit recognition- optimisation results showing recognition error vs iteration.

to SMAC (Fig. 6 (*Left*)). EXP3BO selects digit '2' as that with the highest error (Fig. 6 (*Right*)). The confusion between digits 2, 7 and 4 from shearing and rotation causes comparable performance of other methods.

Our implementation is available at URL https://github.com/shivapratap/ AlgorithmicAssurance_NIPS2018.

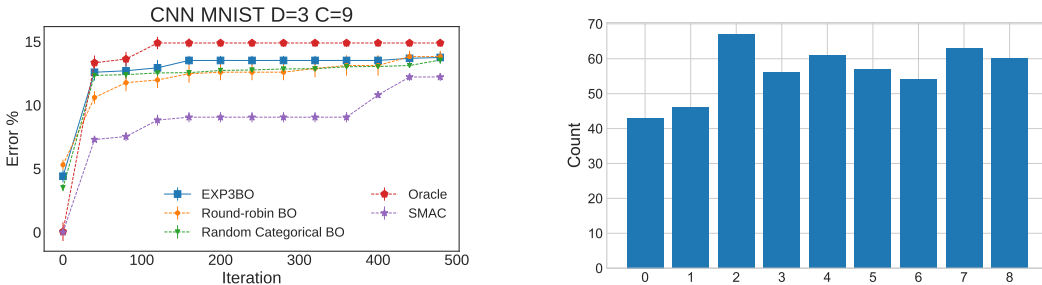

Figure 6 – Multitask assurance using EXP3BO on digit recognition. *Left*: Performance comparison of recognition error vs iterations*; Right*: Histogram of digits selected.

## 5 Conclusion

We have introduced a novel problem of *algorithmic assurance* to assess the deviation of an algorithm from its intended use. We have developed an efficient framework for algorithmic testing for single-task and multi-task settings. The usefulness of our framework is demonstrated on two problems: prediction of strength-determining phases in alloy design and recognition of handwritten digits under shear and rotation distortions. In the modern era of artificial intelligence, algorithms are increasingly taking decisions pertinent to our life, it is very timely to build the confidence that algorithms can be trusted and our proposed algorithmic assurance framework is an early attempt towards this goal.

**Acknowledgements**

This research was partially funded by the Australian Government through the Australian Research Council (ARC). Prof Venkatesh is the recipient of an ARC Australian Laureate Fellowship (FL170100006).

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
