[Reviews · NeurIPS 2018]

Reviewer 1



This paper introduces the problem of algorithmic assurance where they want to systematically check if a (supervised) machine learning algorithm is not deviating from their "true goal" (approximating a function given by a set of observations). The authors propose to use Bayesian optimization to automatically find the input locations (examples/decisions) where the maximum deviation occurs, which means that the overall system won't be producing larger errors. The authors justify their approach and also extend it to "Multi-Task Algorithmic Testing" where they combine this BO problem with a multi-arm bandit setting, developing a Hedge-Bayesian optimization algorithm. The abstract and introduction of this paper are very strong --- probably the best of my batch! Unfortunately, I didn't get too excited about their approach since it's mainly a combination of well-known algorithms. So, my overall score is really a mix of these two feelings. Considering the quality and clarity of this manuscript, I do think this paper can be accepted. But, with respect to the amount of novelty and the impact of this work on a very research-focus community as NIPS, I'm tending to say 'week reject'. Maybe, a small section (or paragraph) of related work will help the reader to appreciate the difference from your analysis to the current research in MAB. minors: line 91: constructing? line 153: x_0being ----- After the discussion and author's response: I think the innovative application of this paper is the strongest feature of this paper. And after reading the author's response I think the main research contribution is also reasonable. I would stress however that Theorem 1 is not formal or particularly interesting. Making it an illustrative example is definitely a good idea. My final recommendation is weakaept.

Reviewer 2



This paper presents a method to find the highest divergence between an ML model their goals by mapping the problem to a Bayesian optimization problem. The idea is very original and I found the paper very creative. The text is easy to follow. Theorem I is not properly defined. The proof is based on an example, which the authors claim can be easily generalized, but they do not provide such generalization. In fact, for that given example, their reasoning is limited to certain stationary kernels. With a general kernel, like a nonstationary kernel or a periodic kernel, there is no guarantee that the narrow peak is avoided. In fact, there are previous works in the literature which explicitly have to add extra features to the standard BO to avoid such optimum, because standard BO will eventually find them. See for example [A]. The combination of a hedging algorithm with BO is not new. It was previously used in [B] to find the best acquisition function. Although the use of hedging algorithm and the application is different in this paper, due to the resemblance, it should be mentioned. In fact, the work of [B] had some issues to seems to be reproduced here, about the hedging algorithm not exploring enough. Concerning the exploration, the experiments show how the multi-tasks scenarios are biased towards a single "task". This is specially surprising in the MNIST case, where there are many confusions between 1 and 7 or between 4, 5 and 6. In fact, 3 is confused with 8 or 5, but those are not explored as much. Also for the MNIST problem, it is surprising that, in the single task scenario, the most difficult setup is where the digits are barely distorted (almost no displacement in X or rotation). Finally, the authors should address how, in their experiments, the round-robin BO is faster and the final results are competitive with the Hedge BO. Why is that happening? [A] José Nogueira, Ruben Martinez-Cantin, Alexandre Bernardino and Lorenzo Jamone (2016) Unscented Bayesian Optimization for Safe Robot Grasping. In Proc. of the IEEE/RSJ Int. Conf. on Intelligent Robots and Systems. [B] Matthew Hoffman, Eric Brochu and Nando de Freitas. Portfolio Allocation for Bayesian Optimization. 27th Conference on Uncertainty in Artificial Intelligence (UAI2011) --- Given the authors response, I think this work might be worth for the NIPS community as a preliminary resource in an interesting line of research.

Reviewer 3



The paper addresses the question: how can we be sure that an ML system behaves as intended. A straight forward method for this assessment is presented, some fundamental properties of the solution are discussed, and the method is tested exemplarily on two real word tasks. Addressing the reliability of ML solutions is, in my opinion, a very important topic. I believe that the discussion will gain intensity. This contribution provides one systematic way to address one of the questions, concerning reliability. As the authors claim---the paper presents "an early attempt towards this goal.". The paper is well written and organized. Still there seem to be a few textual errors, like page 2 line 45-46 "shows that it common to", page 4, line 188 "we use Hedge algorithm", and page 7, line "whose % compositions". Starting on page 7, line 252 there are blanks before the % sign. The general rule is that no blank is used between a number and the percent sign, as is correctly written on page 8, line 294. In addition I assume, that for the final version, the missing capitalization in the literature will be fixed, as in ". hedge", or "gaussian", "bayesian". On page 1, line 13 is written "building trust between humans and machines". This sounds strange to me. I think it is all about the trust, that humans can have in machines, and I do not think that the trust a machine might or might not have in humans is addressed here. Same for page 3, line 95. There is another question I'd like to ask. I'm not sure if the question or the answer to it is fitting in the text, but I ask anyway: If the process of algorithmic assurance makes more effort and needs more expensive labeling by the expert as the initial training of the ML solution, is this something that cannot be avoided, something that we need to accept if we want to get this assurance? And what about the data gather by the process of algorithmic assurance? We could use this data to improve the initial ML solution. Do we not consider this, because this line of thought will only lead us in a recursion. It might be just like not using a validation set for training. It might also be possible, that one could asses the quality of the initial ML solution and then use the new data to train a second, improved ML solution, and then, without assessing the new ML solution, assume that the error is no bigger than the error of the initial ML solution. Young topic, many questions. Post Rebuttal =========== The Author Feedback has sufficiently clarified my questions.